# Beyond Diagnosis and Comorbidities—A Scoping Review of the Best Tools to Measure Complexity for Populations with Mental Illness

**DOI:** 10.3390/diagnostics14121300

**Published:** 2024-06-19

**Authors:** Grace Kapustianyk, Anna Durbin, Ali Shukor, Samuel Law

**Affiliations:** 1St. Michael’s Hospital, 17th Floor, 30 Bond Street, Toronto, ON M5B 1W8, Canada; 2MAP Centre for Urban Health Solutions, Li Ka Shing Knowledge Institute, St. Michael’s Hospital, Unity Health Toronto, 209 Victoria Street, Toronto, ON M5B 1T8, Canada; 3Department of Public and Occupational Health, Amsterdam University Medical Center (UMC), Meibergdreef 9, 1105 AZ Amsterdam, The Netherlands; 4Department of Psychiatry, University of Toronto, St. Michael’s Hospital, 17th Floor, 30 Bond Street, Toronto, ON M5B 1W8, Canada

**Keywords:** diagnosis, complexity, assessment tool, mental health, measurement

## Abstract

Beyond the challenges of diagnosis, complexity measurement in clients with mental illness is an important but under-recognized area. Accurate and appropriate psychiatric diagnoses are essential, and further complexity measurements could contribute to improving patient understanding, referral, and service matching and coordination, outcome evaluation, and system-level care planning. Myriad conceptualizations, frameworks, and definitions of patient complexity exist, which are operationalized by a variety of complexity measuring tools. A limited number of these tools are developed for people with mental illness, and they differ in the extent to which they capture clinical, psychosocial, economic, and environmental domains. Guided by the PRISMA Extension for Scoping Reviews, this review evaluates the tools best suited for different mental health settings. The search found 5345 articles published until November 2023 and screened 14 qualified papers and corresponding tools. For each of these, detailed data on their use of psychiatric diagnostic categories, definition of complexity, primary aim and purpose, context of use and settings for their validation, best target populations, historical references, extent of biopsychosocial information inclusion, database and input technology required, and performance assessments were extracted, analyzed, and presented for comparisons. Two tools—the INTERMED, a clinician-scored and multiple healthcare data-sourced tool, and the VCAT, a computer-based instrument that utilizes healthcare databases to generate a comprehensive picture of complexity—are exemplary among the tools reviewed. Information on these limited but suitable tools related to their unique characteristics and utilities, and specialized recommendations for their use in mental health settings could contribute to improved patient care.

## 1. Introduction

Starting with and extending beyond diagnostics, the proper understanding of patient complexity is an evolving science that is at the core of improving healthcare quality, affecting patient understanding and engagement, referral and service matching and coordination, outcome evaluation, and system-level care planning [1]. Contemporary conceptualizations of patient complexity can be traced back to Rudolf Virchow’s development of the field of social medicine, which arose in response to the detrimental social, physical, and mental health effects of industrialization in the 19th and early 20th centuries [2]. The science of complexity was furthered by advancements in general systemology [3], scientific movement away from reductionism [4], development of integrative medicine [5], as well as George Engel’s model of biopsychosocial medicine in the 1970s [6]. Furthermore, advances in complexity theory offer sophisticated insights and frameworks that contribute to the conceptualization and understanding of patient complexity [7,8,9]. The properties and features of complex systems, such as adaptation, nonlinearity, interactive causal structures, emergence, self-organization/spontaneous order, and feedback loops, provide insights into mental health and illness, particularly in relation to the dynamic context of how individuals and systems respond to stresses and changes [10,11]. With concepts like the “edge of chaos”, which explores complex adaptive coping with chaotic environments through flexibility, agility, and innovation, fostering adaptability and resilience, chaos theory also contributed to the understanding and development of complexity measurements [12].

Researchers and clinicians have built on some of these concepts and developments to arrive at a variety of tools that enable a more rigorous measurement and assessment of patient complexity, becoming increasingly multidimensional and multidisciplinary in perspective. Early concepts of the complex patient were disease-oriented, based largely on the strength of the diagnosis and attendant condition of the illness, plus the presence and number of multi-morbidities; however, this has been progressively recognized as being far too narrow and inadequate in reflecting the realities of complexities [13,14]. In the mid-1990s, the concept and research on complexity moved more fully to include various determinants of health, integrating social, economic, political, and environmental aspects [15].

As a steadily growing field, the definition of patient complexity remains fragmented and, often, quite subjective. For example, a review of the literature for definitions and descriptions of complexity found three broad themes: multimorbidity (including the key diagnosis), resource utilization, and psychosocial complexity, with multimorbidity being the formative and most investigated term [1]. In this category, most measures on complexity still emphasize chief diagnosis, physical health issues, and comorbidity (e.g., the Charlson Comorbidity Index [16], the Elixhauser Comorbidity Index [17], and the Johns Hopkins’ Adjusted Clinical Group system [18]), and largely exclude social determinants of health, despite overwhelming evidence of their relationship with health outcomes [19]. The resource utilization category takes up complexity as an outcome of resource utilization, impact, and efficiency (e.g., the Kaiser Permanente Chronic Conditions Management pyramid [20]), while the psychosocial category emphasizes some combination of psychosocial and environmental factors that include social isolation, psychiatric illness, socio-demographic vulnerability, addictions, and access to care, among others. Given these various definitions of complexity, researchers have proposed models that are based on the chief diagnosis, but extend beyond this to include various aspects of a patient’s health status as vectors in a multidimensional space, with each of the important dimensions represented. Analyzing the relationships and interactions within this vector space helps to capture the overall complexity. Some vector models focus on clinical complexity [21,22], while others highlight the balance between patient workload demands and patient capacity [23] or offer a systematic and holistic perspective to understand care needs and patient challenges to ultimately match services and inform guidelines and policies [1].

Within the field of complexity measurement, populations with mental illness present unique challenges as their diagnoses are typically without any hard anchors of biological markers and rely on reported and observed psychological and behavioral information, using an epidemiologically and statistically driven approach that heavily relies on subject experts [24,25]. The current chief diagnostic references and guidelines are the Diagnostic Statistical Manual (DSM-V) and International Classification of Diseases (ICD-11) [26]. Psychiatric diagnoses and complexity arise from biological, psychological, and social components that are fundamentally more intertwined and interactive; in addition, acute and chronic physical health conditions, addictions issues, and neurocognitive disorders often coexist [11]. Their complexity is multifactorial and is also mediated by patient-level factors, clinician practice factors, systemic-level issues and barriers, and health policy factors [27]. Moreover, social and cultural forces, such as stigma, insight and acceptance of the illness, help in seeking patterns, adherence to treatment, and availability and access to treatment and care, all affect the understanding, approach, and treatment of mental illness. These intersecting challenges often require special understanding and consideration [23]. Furthermore, within those with mental illness, a unique subgroup of people with serious mental illness (SMI), or formerly termed severe and persistent mental illness (SPMI)—mainly consisting of people with challenging diagnoses, such as schizophrenia or psychotic spectrum disorders, major mood disorders such as bipolar disorder and severe depression, often with comorbid addiction disorders—is known to have very high levels of resource utilization as well as poor health outcomes (including non-psychiatric illness, such as cardiovascular disease). According to care guidelines [28,29], people with SMI require intensive multidisciplinary team-based care, with special attention to individualization of services (i.e., person orientation), taking into account the individual’s biopsychosocial context. Improving both resource allocation and health outcomes require sophisticated case-mix adjustment tools, while functions such as team-based care, integrated care, and individualization of care require person-oriented complexity assessment tools [30]. Therefore, proper complexity measurement for people with SMI presents additional challenges.

Given the many recent developments in tools or algorithms to measure complexity, and a lack of attention paid specifically to mental health in the field, we sought to review the *scope* of complexity tools available and consequently determine which could best serve patients within mental health settings, including for those with SMI. Properly conducted, measurements of complexity could aid in mental healthcare through the proper assignment of services based on diagnostic groups, improved admission and discharge decisions, allocation of resources, evaluation of performance and quality of care, and designs of care teams and governance structures—all of which could aid in improving care from a deficit-oriented focus on mental illness to a strength-focused approach in promoting mental health [15]. Adjustments for complexity, or case-mix, are also key to understanding baseline differences in groups when interpreting outcome data and making attribution claims and policy decisions [31]. This review aims to:Undertake a review of the tools that define, operationalize, and measure patient diagnoses and complexity, while being considerate of the fact that there is not one agreed upon definition by complexity practitioners or researchers.Identify tools that could be applied specifically to patients with mental health diagnoses, including those with SMI.

## 2. Methods

A scoping review methodology informed by the PRISMA-ScR (Preferred Reporting Items for Systematic reviews and Meta-Analyses extension for Scoping Reviews) was used to identify tools and measures of patient complexity [32] to determine how patient complexity is defined and operationalized through such tools and to select tools that could be applied specifically to populations with mental illness. Relevant databases (i.e., PubMed, Google Scholar, PsycINFO, and APA PsychInfo, Embase) were searched between March 2020 and November 2023. There were no restrictions on publication date or language. For Google Scholar, the authors limited the search to 1000 results. Key terms searched in combination included: complexity, patient complexity, tools, serious mental illness, severe mental illness, severe and persistent mental illness, and mental health. There were also no restrictions related to symptoms or diagnoses to keep the search as broad as possible.

The authors also searched for reviews with each combination (e.g., mental health AND complexity AND review). In addition, the authors consulted a key review in the literature by Schaink and colleagues [1], and hand-searched the bibliography and future references to this review, particularly for articles and reviews that focused on patient complexity in our target population with psychiatric diagnoses. Inclusion criteria: Articles were included if they substantially engaged in operationalizing a tool for patient complexity that included biopsychosocial domains and were, at minimum, field-tested in a setting or population with mental health diagnoses. This allowed us to extend beyond the diagnoses, conceptual narratives (e.g., diagrams or profiles), and frameworks (e.g., vectors) to find tools that could be used and implemented in healthcare settings. Articles were excluded if the tool related to only one particular dimension but did not address a fuller scope of patient complexity (e.g., multimorbidity or experience with hospitalizations only but lacking any psychosocial determinants data). The objective was to avoid tools that would have only captured the “traditional” understanding of complexity, like multimorbidity or resource utilization, while acknowledging that there is no single agreed upon definition of “patient complexity”. Information on the respective tool’s definition of complexity, aim and utilization, setting/sector intended, geography and reach, study sample and whether the tool was mental health population field-tested or validated, level of validation, primary age group and socioeconomic status (SES) of the sample, tool funding source, and historical complexity tools referenced was extracted and outlined in summary tables.

While not an exclusion criterion established a priori, we prioritized tools that included discrete mental health or psychiatric diagnostic AND psychosocial domains. Additional information on these domains and their relevant sources, outcomes assessed in relation to complexity measured, scoring, and recommendations provided were extracted. We also included additional comments based on the team’s expertise and experience using the tool, such as ease and mode of use, time taken to complete the tool, adaptability (e.g., can one use a portion of the tool to shorten it but still have useful outcomes?), government or national association recommendations, any patient specificity or recommended settings, and whether they are available to use for free and in what languages. This was information often missing in general reviews, and it was extracted with the intention of providing practical information required by users when deciding if, and when, to use any given tool or measure.

Given the nature of a scoping review, there was no required oversight by an institutional review board. The scoping review protocol was registered with Open Science Framework (OSF) at https://doi.org/10.17605/OSF.IO/4QSDG (accessed on 5 November 2023).

## 3. Results

### 3.1. Search Results

The search returned 5345 articles (4727 articles after deduplication) from PsychINFO (741 articles), EMBASE (81 articles), PUBMED (349 articles), Google Scholar (4142 articles) , and hand searching (32 articles published until mid-November 2023). After the initial title and abstract screening, there were 165 articles included for full-text review, of which 14 were considered relevant for the final selection as determined by the inclusion and exclusion criteria. For full-text review, the main reasons for exclusion were not a tool/operationalized (71 articles), not patient complexity (63 articles), not field-tested or validated in a mental health population or setting (15 articles), other language duplicate (1 article), and protocol with no associated discussion paper (1 article). (Please see Appendix A for PRISMA-ScR flow diagram and Checklist.) Some excluded papers that proposed frameworks for patient complexity or contributed in discussions regarding aspects of complexity—such as multimorbidity, level of care, continuity of care, quality of life, and self-sufficiency (e.g., the Dutch Self-Sufficiency Matrix) [33]—were reviewed for their content and are referenced in this review to capture the breadth of complexity and enrich the discussion.

### 3.2. Description of the Final Complexity Tools and Papers Reviewed

The 14 final complexity tools and related papers, along with all the extracted details for the respective tools, are presented in Table 1a,b. There was considerable variety in complexity definition, measurement indicators, setting, and use. While all included tools captured mental health diagnoses and complexity, only eight included a definition of complexity that was used to guide their tool development, ranging from vague (e.g., COMPRI [34]: “Complexity of care”; and OCCAM [35]: “The number of different factors that affect the illness and its management”) to more comprehensive definitions (e.g., VCAT-CM [30]) that used a multidimensional person-oriented profile, AMPS [36], and MCAM [37] that recognized medical and non-medical factors that interfere with care and improved health; and PCAM [38] that included “Social determinants of health that characterize socioeconomic disadvantage…, which lead to a complex interplay of biological, psychological, and social factors”.

The geographic context in which each of these tools was developed was also wide-ranging. For examples, VCAT [30], HexCom [39], Escalation tool [40], and OCCAM [35] were developed for specific geographic regions in Vancouver (Canada), Catalonia (Spain), Sydney (Australia), and Oxford (England), respectively, while other tools were developed for national (e.g., PCAT [41]—United Kingdom) or multinational use (e.g., COMPRI [34]—Spain, Italy, Hungary, Netherlands, Portugal, Germany, Denmark). Importantly, these tools were also developed for a myriad of healthcare settings and uses. The Care Process Model (CPM) [42] was developed for the general patients of the Utah’s Intermountain Healthcare Hospital system in the US to determine which level of care best suits the patient; the Minnesota Complexity Assessment Model (MCAM) [37] was developed for clinicians to quickly articulate and take account of “unnamed medical factors” that interfere with care delivery in that US state; the Complexity Checklist [43] was developed to discriminate between patients of Primary Care Providers (PCPs) that did and did not require complex care in the Colorado Division of Mental Health. Finally, the INTERMED [44], the formative complexity tool developed in the Netherlands, which many tools in this review are adapted from or informed by, has been further adapted and validated for use in multiple settings, including general healthcare, mental healthcare, and those with multiple diagnoses (substance use disorders and physical and mental disorders) (see Table 1a,b for a complete overview of all the above characteristics of each tool).

**Table 1 diagnostics-14-01300-t001:** (**a**) Characteristics of the final tools selected. (**b**) Characteristics of the final tools selected (continued).

(a)									
First Author	Year	Title	Tool	Definition of Complexity	Tool Aim	Tool Utilization	Study Sample	Geography and Reach	Intended Setting or Sector
			*Name of the tool developed*	*What is the definition of complexity used by the authors for tool development?*	*Why was it created?*	*How the tool is used (e.g., discharge vs. service utilization/redirection)*	*Who is included in the paper’s study sample?*	*Rural, urban, or comprehensive?*	*What was the tool’s intended health setting?*
Health Connection [36]	2014	HealthConnection clinic complexity assessment tool (AMPS): an introduction; User Guide	The Complexity Assessment Tool (also known as AMPS—Attachment, Medical, Psychiatric, Social)	Used to help to identify the medical and non-medical factors that interfere with care and improved health	AMPS was integrated into the Health Authority’s EMR, providing a standard that enables providers to assess patient complexity, guide attachment to providers, and to develop individualized care plans	Service utilization and care planning	“Highly complex” patients often with a history of challenging patient–provider relationships (the “over-serviced but underserved”)	Vancouver, Canada; Urban	HealthConnection Clinics in Vancouver, BC, Canada
Busquet-Duran [39]	2020	Describing Complexity in Palliative Home Care Through HexCom: A Cross-Sectional, Multicenter Study	The Hexagon of Complexity (HexCom)	“Gap between patient needs and healthcare services”, or “A mismatch between patient needs and services” (situations that are refractory to treatment options defined as “high complexity”; situations that are difficult to resolve defined as “moderate complexity”	To describe differences in complexity across disease groups in specific home care for advanced disease/end-of-life patients, both in general and relating to each domain and subdomain	Distinguish between those who need specialized palliative care and those who do not	Patients with advanced disease and/or at end of life attended by palliative care teams at their home	Catalonia, Spain	Patients at end of life in home care in Catalonia, Spain
Carpenter [40]	2021	The development of pathways for responding to patient complexity in a liaison psychiatry setting	Escalation Tool	No info available	To identify complexity in general hospital inpatients and guide pathways for action	Pathways for step-wise escalation of response	Consultation liaison psychiatry patients	Sydney, Australia; Urban	Consultation liaison psychiatry services within general hospital care
de Jonge et al. [44]	2005	Operationalization of biopsychosocial care complexity in general healthcare: the INTERMED project	INTERMED Complexity Assessment Grid (INTERMED)	“The identification of biological, psychological, social and health system factors considered interacting in health complexity”	To address the issue of how to approach biopsychosocial complexity in general healthcare (systematizes a biopsychosocial approach to ascertain case complexity)	Case-mix decision support and outcome management	Based on patients admitted to a general medical ward with somatic illnesses	Validity study: Amsterdam, The Netherlands; Urban Reliability study: Switzerland; Unclear	General healthcare
Hudon [45]	2021	CONECT-6: a case-finding tool to identify patients with complex health needs	COmplex NEeds Case-finding Tool—6 (CONECT-6)	Based on the multiple chronic conditions research definition: “The gap between an individual’s needs and the ability of health services to meet those needs”	To develop and validate a rapid (less than 2 min), self-administered 6–8-item (yes or no answers) case-finding tool to identify patients with complex health needs	Case-finding	Adults with three or more visits to the ED within 12 months and that presented at least one severe medical illness (e.g., cardiovascular and seizure)	Quebec, Canada; Unclear	Emergency departments across the province of Quebec, Canada
Huyse [34]	2001	COMPRI—An Instrument to Detect Patients With Complex Care Needs	Complexity Prediction Instrument (COMPRI)	Not specifically defined	To improve the detection and treatment of patients with combined medical and psychiatric problems	To identify complex patient groups that could benefit from integrated longitudinal coordinated care, including case management	Patients admitted to 1 of 11 general internal medical wards from 7 European countries	Europe (Spain, Italy, Hungary, Netherlands, Portugal, Germany, and Denmark); Unclear	General hospital care
Martin-Rosello [46]	2018	Instruments to evaluate complexity in end-of-life care	IDC-PAL	“…related to the clinical situation, to the person and their family, requiring a prior multidimensional assessment by the multi-professional team; and related to the intervention scenario, including from the professionals and healthcare systems, to the community, requiring a broader multi-referential approach”.	To support and help to coordinate professionals involved in end-of-life care and to maximize consensus among professionals of the different level-of-care provision, facilitate effective communication between resources, and enhance a shared care model for palliative care	Care coordination for patients in palliative care	Patients from healthcare centers with palliative care services	Andalusia, Spain; Unclear	Palliative care services
Mount [47]	2015	Patient care complexity as perceived by primary care physicians	Complexity checklist	Did not define explicitly; operative definition: “…patients were generally considered complex based on the adverse impact on their practice… and excessive encounter times”	To discriminate between patients in clinical practices who did and did not require complex care	To improve the care of complex patients in primary care and to improve the confidence and capacity of primary care providers	Clients identified by primary care providers as complex	County in a Northwestern State, United States; Unclear	Primary care providers
Oniki [42]	2014	Computerization of Mental Health Integration Complexity Scores at Intermountain Healthcare	Intermountain Healthcare’s Mental Health Integration (MHI) Care Process Model (CPM)	None included	To determine which of the three levels of care is appropriate for the patient	Care planning for the level of care and resources needed	General patients of the Intermountain Healthcare Hospital system	Utah, United States; Unclear	Mental health in a primary care setting
Peek et al. [37]	2009	Primary Care for Patient Complexity, Not Only Disease	Minnesota Complexity Assessment Model (MCAM)	“The person-specific factors that interfere with the delivery of usual care and decision-making for whatever conditions the patient has”	To provide a simple vocabulary and method for clinicians to quickly articulate and take into account what are often seen as diffuse and unnamed “nonmedical factors” that interfere with delivering care, interfere with obtaining the expected results, and create the sense of being “stuck”	Care delivery/service redirection; care planning	None—face validity is ascertained using vignettes	Minnesota, United States; Unclear	Fast-paced primary care settings
Pratt et al. [38]	2015	The Patient Centered Assessment Method (PCAM): integrating the social dimensions of health into primary care	Patient Centered Assessment Method (PCAM)	“Social determinants of health that characterize socioeconomic disadvantage lead to a complex interplay of biological, psychological and social factors—the impact of these various characteristics is conceptualized as ‘patient complexity’”	Developed as a Keep Well anticipatory health-check screening tool to integrate the social dimensions of health, including mental health checks into primary care practice	To identity biopsychosocial complexities in a manner that facilitated referral to the appropriate medical, lifestyle, psychological, social, and self-help services in a more effective way	1. Primary care clinics offering additional KeepWell services for people at risk of cardiovascular disease. 2. Nurses working with complex patient populations (i.e., homeless, refugee, and travelling communities).	Scotland; Unclear	“Keep Well”—primary care settings targeting cardiovascular disease and diabetes risk identification and reduction in highly socioeconomically disadvantaged settings.
Shukor et al. [30]	2019	A Multi-sourced Data Analytics Approach to Measuring and Assessing Biopsychosocial Complexity: The Vancouver Community Analytics Tool Complexity Module (VCAT-CM)	Vancouver Community Analytics Tool Complexity Module (VCAT-CM)	Multidimensional person-oriented profile comprising the nine domains, which are measured as vectors (i.e., having magnitude and direction)	Patient care could be strengthened if measurement use is complemented with person-oriented knowledge synthesized from other existing databases and sources	Development of real-time person-oriented biopsychosocial complexity profiles to enable community health centers to operationalize the fundamental building blocks of primary care	Patients of the Vancouver Community Health’s Raven Song Community Health Centre	Vancouver, Canada; Urban	Vancouver Coastal Health’s Community Health Centre clients
Troigros [35]	2014	Measuring complexity in neurological rehabilitation: the Oxford Case Complexity Assessment Measure (OCCAM)	Oxford Case Complexity Assessment Measure (OCCAM)	Complexity relates to the number of different factors that affect the illness and its management	Part of a service development process aiming to improve costing and to understand outcomes better	Service development	Patients receiving rehabilitation after acute onset disability with various neurological diseases, including stroke, traumatic brain injury, spinal disorders, multiple sclerosis, and cerebral hypoxia; in- or out-patients.	Oxford, United Kingdom; Unclear	Specialist neurological rehabilitation service
Turner-Stokes [41]	2019	The patient categorisation tool: psychometric evaluation of a tool to measure complexity of needs for rehabilitation in a large multicentre dataset from the United Kingdom	Patient Categorisation Tool (PCAT)	None specifically defined	Originally developed as a checklist to identify patients with complex needs requiring treatment in tertiary inpatient rehabilitation care—then developed as an ordinal scale to identify patients with difference complex levels	To identify the complexity of the clinical caseload across different services and to signpost services to the different levels, with appropriate streams	Multi-center cohort of patients from the national clinical dataset representing 63 specialist rehabilitation services across England	England; Unclear	Patients presenting for specialist neurorehabilitation
**(b)**									
**First Author**	**Year**	**Population with Psychiatric Diagnoses Feld-Tested or Validated**		**Level of Validation**	**Available Tool Psychometrics**	**Primary Age Group**	**Primary SES of Population**	**Tool Funding and/or Insurance Sources**	**Historical References**
		*Is it validated in a specific mental health setting or preferably in a SMI/SPMI population?*		*Is it validated or field-tested?*	*Sensitivity; specificity; reliability, etc.*	*What is the primary age group that the tool was developed for or tested in?*	*What is the primary social economic status group that the tool was developed for or tested in?*	*How is the tool funded?*	*Does the paper or tool cite or adapt previous complexity tools? If so, what are they?*
Health Connection [36]	2014	Yes—only piloted in this population * * Author correspondence, late 2021		Piloted with a heavy focus on psychiatric, mental health, and addiction populations	No known info	Not stated	Not stated	Funded by Vancouver Coastal Health	MCAM
Busquet-Duran [39]	2020	No—palliative home care settings		Partial validation	Reported high inter-rater reliability (Kappa = 0.92)	No specific age; reported mean age = 78.7 years (SD = 13.0), range = 22–107 years.	Not stated	The research institute (IDIAP Jordi Gol) funded the databases, plus an internal grant from the Metropolitan Nord Primary Care Service (Catalan Health Institute)	Multiple Chronic Conditions Research Network
Carpenter [40]	2021	Yes—Consultation Liaison Psychiatry setting		Field-tested over a 2-week period	No known info	Not stated	Not stated	Not stated	INTERMED, PCAM
de Jonge et al. [44]	2005	Yes— validated in many settings, including a somatoform/specialized mental health outpatient setting; those with triple diagnoses (substance use disorders and physical and mental disorders)		A combination of psychometrics and clinimetrics available. Reported face validity and ease of use in multiple care settings globally.	Reliability: (pooled data): Cronbach’s alpha = 0.78–0.94. Sensitivity: ranging from 0.58 (internal medicine) to 0.94 (low back pain). Specificity: ranging from 0.45 (low back pain) to 0.94 (multiple sclerosis).	Not stated	Not stated	Not stated	Iterations of INTERMED
Hudon [45]	2021	No—emergency departments only		Initial validation available	Sensitivity: 90%—for a threshold of two or more positive answers. Specificity: 66%—for a threshold of two or more positive answers.	Adults ≥ 18 years old. Mean age of participants = 67 years (SD = 20.0).	Not stated	Not stated	INTERMED; Multiple Chronic Conditions Research Network
Huyse [34]	2001	Unclear—population has combined medical and psychiatric problems, but does not specify any criteria or diagnoses required.		Predictive validity and reliability available	Reliability: 0.55 Pearson correlations for different complexity indicators and combinations for mental health problems.	Mean age = 62.1 years (SD = 17.2)	Not stated	Not stated	INTERMED
Martin-Rosello [46]	2018	No—across palliative care services in general		Reported content validation, reliability, and field-tested at the national level, but the publication of results pending.	No known info	No specific age for palliative care stated.	Not stated	Not stated	Hui’s criteria, PALCOM, INTERMED
Mount [47]	2015	Yes/unclear—not validated in a specific mental health setting, but 58% of patients categorized as complex (class 4), comprising patients who have mental health issues, multiple diagnoses, and poor follow up		Dimensions of complexity tool were validated, but not the tool itself. Face validity based on providers’ subjective sense of complexity	No known info	No specific age; the average range in the analysis was 50–59 years for complex patients	Not stated	Not stated	MCAM, INTERMED, PCAM
Oniki [42]	2014	Yes/unclear—patients with mental health issues in a primary care setting but uncertain if the tool was tested in a population whose mental health diagnosis is the primary diagnosis; preliminary analysis involved patients with suicide crises.		Overall, the diagnostic algorithm has not yet been validated. Some sub-score analysis reportedly published internally.	No known info	Not stated	Not stated	Not stated	None
Peek et al. [37]	2009	No—but involved individuals with a mental health condition “in the mix”		Field-tested—with periodic feedback and suggestions from family medicine faculty members and additional language and method validation by individual faculty, small care teams, and medical residents	No known info	Not stated	Not stated	Not stated	INTERMED
Pratt et al. [38]	2015	No—tested in heart disease, high needs high care, home care, and primary care populations with no details on psychiatric diagnoses		Face validity and preliminary external validity testing via a qualitative exploratory study on tool applicability, acceptability, and feasibility	No known info	Study 1: mean age = 54 years (SD = 6). Study 2: mean age = 52 years (SD = 12).	1. Unclear. 2. Likely low-income given low housing status and potential work status.	Healthier Scotland, a division of the Scottish government	MCAM, INTERMED
Shukor et al. [30]	2019	Yes—Community Health Centre setting where 99% of the population have at least one mental disorder		Face validity assessed by physicians at the individual client level and by the medical director at the client population level.	No known info	Not stated	Low-income, food insecure, housing insecure or homeless and face difficulties associated with access to social and healthcare services	Vancouver Coastal Health—one of six publicly funded Regional Health Authorities in British Columbia, Canada.	AMPS, MCAM, INTERMED
Troigros [35]	2014	No—neurological rehabilitation population		Validated—in the absence of a ‘gold standard’; concurrent convergent and discriminant validity assessed through Spearman correlations with INTERMED, the Rehabilitation Complexity Scale, and team judgement scale.	Reliability: Inter-rater-weighted K = 0.85, *p* < 0.001; Cronbach’s α coefficient = 0.69 Sensitivity: 84.6%, for optimal cut-off ≥34 Specificity: 62.8%, for optimal cut-off ≥34	Mean age = 51.1 years (SD = 17.1)	Not stated	Not stated	INTERMED, the Rehabilitation Complexity Scale
Turner-Stokes [41]	2019	No—rehabilitation population, referenced in traumatic brain injury and -spinal cord injury studies		Structural validity tested with the multi-center cohort of patients from the national clinical dataset representing 63 specialist rehabilitation services across England. Concurrent and criterion validity tested through a priori hypothesized relationships with other validated measures.	Sensitivity: Category A (less complex)—76%; Category B (more complex)—85% Specificity: Category A (less complex)—75%; Category B (more complex)—78%.	Mean age = 54.4 years (SD = 18.2) for the total sample (but study setting catered to predominantly working-aged adults (16–65 years)	Not stated	Not stated	Rehabilitation Complexity Scale (RCS-E), the UK Functional Assessment Measure (UK FIMþFAM), the Northwick Park Nursing Dependency Scale (NPDS)

### 3.3. Complexity Tools Useful for People with Mental Health Diagnoses and Challenges

Six of the fourteen tools met the specific focus of this review as they were field-tested or validated in a mental health setting—AMPS [36], CPM [42], INTERMED [44], VCAT [30], Escalation tool [40], and Complexity checklist [43]—of which four were directly developed for and tested in a mental health setting (AMPS, INTERMED, VCAT, and Escalation Tool) and two (Complexity checklist and CPM) had strong representation or identified individuals with mental illness and psychiatric diagnoses in a general healthcare setting. Each of the six tools has unique attributes that define them, and these attributes help to guide end-users in their decision on which tool(s) to use based on their unique needs. These attributes—namely the domain types and domain-related data sources, domain and overall tool scoring, and recommendations provided on scoring—all focus on different key aspects of patient complexity and are described and analyzed in depth below. A summary is presented in Table 2a,b.

*Domain types*: While each tool contains at least one specific psychological domain (e.g., “Psychological”, “Suicide assessment”, and “Medical, addictive, and psychiatric comorbidity” (i.e., multiple diagnoses permitted)), only the INTERMED, Escalation Tool, VCAT, and AMPS had separate, comprehensive psychosocial domains. These additional domains ranged from “family and living situation”, “previous complexity or escalation”, and “activities of daily living” to “risk of harm”. Of note, no tool had a specific race, ethnicity, or inquiry on experiences of racism domains.

*Domain sources*: Most tools (INTERMED, Escalation Tool, AMPS, and Complexity Checklist) are all primarily scored from the clinician’s perspective, while the VCAT and CPM use multiple domain-specific sources that include a combination of hospital admission records, patient/clinician perspectives, and validated questionnaires, like the Patient Health Questionnaire-9 (PHQ-9) [48] or the Health of the Nations Outcome Scale (HONOS) [49].

*Domain and tool scoring:* The VCAT, INTERMED, CPM, and AMPS all have discrete scoring with summary scores and sub-scores available, whereas the Escalation Tool and Complexity Checklist are checklists that PCPs used to determine the levels of care needed (e.g., if a PCP checked off three or more domains that they believe contributed to the patient’s complexity, the patient is escalated to a higher level of care).

*Score-based recommendations from the tool*: Dependent on the overall tool scores or thresholds surpassed, the tools may provide recommendations on the next steps and action items for the patient and/or practitioner. For example, the CPM recommends a care manager and mental health specialist in addition to routine, PCP-based care for those with a higher complexity, while the INTERMED simply flags a higher complexity for consideration of higher need for care. Of note, the VCAT’s summary score was initially used to gauge whether the existing patients were meeting the mandate of the services provided, but later leveraged the scores to decide on access to team-based care within community-health centers.

*Additional considerations*: All but the CPM and VCAT are pencil and paper tools, available within the published papers; the others are computer-based. There is little information about licensing or potential costs associated the tools. Most tools have received positive feedback on their feasibility and use. The INTERMED is available in nine languages for the self-assessment version, while the other five currently appear to be in English only. The AMPS, INTERMED, and Complexity Checklist have versions that are available online. Only the Complexity Checklist and INTERMED papers report the time taken to complete the tool, which are an average of 1.4 min and 20 min, respectively (see Table 3 for details of these additional considerations).

### 3.4. Complexity Tools for People with SMI

People with SMI diagnoses have a diverse set of complexities and unique clinical and psychosocial challenges. With this in mind, the INTERMED [44] and VCAT [30] were identified as the best suited tools as they include both psychological and social domains, represent both clinician-scored and multiple-sourced tools, are field-tested or validated in a mental health setting, and have varying levels of adaptability required. More specifically, the INTERMED is a four domains-by-three time perspectives grid approach directly rated by clinicians. The four domains are *biological*, *psychological*, *social*, and *healthcare*, and the three time perspectives are *history*, *current state,* and *prognosis*. The 4 × 3 grid approach generates at least 12 specific complexity data categories (e.g., *biological* x *history* may include intensity of prior treatment, etc.), and each of the 12 categories can be subdivided into finer points of information (e.g., *social* x *current State* can include residential instability and impairment in social integration, etc.).

On the other hand, the VCAT is a computer-based instrument that utilizes information from healthcare databases and clinician input to generate a comprehensive picture of complexity. The VCAT has nine domains, conceived as vectors arrayed in parallel, and they form a profile of complexity rather than a singular index. The nine domains include quality of *attachment* to care; *service density* in terms of multitudes of care and their access and coordination; *social and environmental factors* related to social barriers, such as housing instability, poverty, etc.; *psychosocial factors,* such as cognitive, behavioral, and/or functional impairment; *relationships* issues, such as inability to maintain lasting personal or professional relationships; the functional level of *activities of daily living (ADLs); medical complexity,* such as chronic disease, concurrent disorders, or communicable diseases; *acute (hospital) utilization,* such as emergency department use or hospitalization; and *Risk of harm to self or others* [30]. VCAT leverages routinely collected administrative and clinical data to support more consistency and reliability in data collection as well as having potential for larger level scalability across the system and for program evaluations.

Although a small suite of biopsychosocial complexity tools derived from the Dutch INTERMED tool address biopsychosocial complexity (MCAM, PCAM, etc.), item subjectivity and reliance on additional data collection are key barriers to their widespread use.

## 4. Discussion

This current review aims to contribute to the understanding of patient complexity tools available, with a special emphasis on those with psychiatric diagnoses being served in mental health settings, in view of the limited definitional clarity surrounding this topic and the wide-ranging developments and available choices. Our findings present an overview of the patient complexity tools that specifically include psychological and psychosocial domains and are operationalized for use in mental health settings, with additional attention to those that are suitable to people with SMI who have unique challenges. Despite the progress in concept, it is evident from this review that the number of currently available complexity measuring tools that consider a broader set of biopsychosocial aspects of complexity is still limited, and fewer still are best suited for populations with mental health challenges. Encouragingly, the review did identify a few readily available, useful tools that could serve those with a wide spectrum of psychiatric diagnoses and illnesses.

Previous major reviews of complexity measurement have focused on how complexity is defined [16] or thematically classified [1], what conceptual frameworks of medical complexity are available [50], provider narratives on complexity [14], or how comorbidity does not fully reflect complexity [51]. While each of these reviews had merits and examined some key components for measuring complexity, they did not offer a comprehensive framework that informed the state-of-the-art measurement of complexity for mental health settings. Bearing in mind that no single universal definition for complexity exists, the current review addressed these limitations by purposefully identifying a broad range of perspectives, tools, and their respective strengths and carried out literature-informed data extraction and comparisons. The current review potentially contributes to making complexity measurement for people with mental illness more understandable, more accessible, and more widely utilized, which could in turn improve mental health outcomes across various contexts.

One of the findings from the review is the high level of heterogeneity, even among the tools that met the inclusion criteria. Given the diversity both of the definitions of complexity and the contexts under which each of the tools was developed, it is not surprising that the tools differ a great deal in their orientations and foci. Only one (i.e., INTERRMED) is specifically adapted for people with mental illness [44]. The other few that were found to be suitable for people with mental illness have more explicit psychological and/or psychosocial domains, but they may not be specifically developed for this population. At the least, the current review is helpful in understanding how such heterogeneity came about and realizing where the limits are and could inform further specific developments in this field for people with mental illness.

Of the six complexity tools for patients with mental illness identified—the AMPS [36], CPM [42], INTERMED [44], VCAT [30], Escalation tool [40], and Complexity checklist [43]—the current review summarized and compared highly specific details on the characteristics and indicators of each tool to assist with decision-making processes when selecting a tool, depending on one’s needs, population, and resources available. The review found that, while each of the six tools included a psychological domain, only four (i.e., the AMPS, INTERMED, VCAT, and Escalation Tool) also included a psychosocial domain. For end-users interested in a more holistic approach, the current review would endorse these four, as they have more mental health-inclusive considerations through additional domains, such as family and supportive relationships, living situation, financial capacity, and history of substance abuse, among others. The current review is explicit in advocating the inclusion of such important psychosocial issues in understanding a mental health patient’s complexity, as the field has come a long way from the days of the more unidimensional consideration of patient complexity [16].

The current review also shows that the more recently developed, comparatively more sophisticated, biopsychosocially informed tools—such as the VCAT and INTERMED—cover the various perspectives and address the shortcomings of the previous tools by having good “cross-mapping” with many of the specific dimensions of the other more limited complexity tools (e.g., the MCAM [37], AMPS [36], etc.). However, to further improve complexity measurement, some experts have advocated capturing better the interrelationship among the domains [52] and to not only focus on the deficits involved in complexity, but consider wellbeing and well functional status too [53]. In addition, it is very important that concepts from complexity theory—such as nonlinearity, interactive causal structures, and the “edge of chaos” phenomenon from chaos theory—be further explored and incorporated into the conceptualization, development, and refinement of complexity tools [8,12].

A more holistic understanding of complexity has substantial implications for patients, physicians, administrators, policymakers, and insurers alike. Some researchers pointed out that, for example, if the demanding care of complex patients is not appropriately considered and compensated, and healthcare provision are not harmonized with complexity, service providers have a powerful incentive to select healthier patients and neglect those who are more complex [54]. As a field, primary care has demonstrated that proper complexity measurements have contributed to improvements in provider–patient communication, service empanelment, team-based care, service matching, coordination and continuity of care, and outcome research [55,56,57]. Mental health as a specialty, being less exposed to such care improvements through incorporating complexity measurements, may find it attractive to adapt such measurements [42,58].

In terms of access and ease of implementation, four of the six mental health compatible tools identified (i.e., the INTERMED, AMPS, Complexity checklist, and the Escalation tool) rely on the clinician’s perspective. This approach does carry compromises on the aforementioned need for comprehensiveness as well as the usual limitations of inefficiencies and subjectivity of questionnaire-based methodology. On the other hand, as they do not require administrative or other sources of data, they have a low barrier of use. Also, these tools also allow the clinicians to make real-time specific recommendations for further action, making them more attractive with immediate utility, and the ease of use could mean a higher level of utilization and potential impact [59]. Of the two that need administrative and other clinical data, such as hospital admission records or electronic health records. (i.e., the INTERMED and VCAT), the INTERMED does offer a self-assessment version as well, making it more accessible. The VCAT, while harder to use, shows promise as a very powerful tool with mounting evidence of adaptability [30,60]. If fully realized, the VCAT developers envision that it can achieve—from clinical to organizational governance levels—improvements related to assessing whether certain patients meet the service organization’s mandate, optimizing and balancing the client panels of healthcare providers, optimizing the composition and organization of multidisciplinary teams, assessing workload content, enabling the recognition of client needs and the tailoring of individualized care plans, and the monitoring and assessment of changes in individual and population complexity profiles and outcomes over time [30]. It is hypothesized that complexity profiles or scores are quite sensitive to change and possibly able to identify or predict clinical presentations (e.g., the future onset of frailty in aging populations) based on multitudes of relevant information (e.g., relationships, ADLs, disability status, substance use, etc.) [30]. In practical ways, such utility in mental health settings could mean a better matching and prioritizing of patients who require intensive community outreach services, such as Assertive Community Treatment teams, which are always on short supply and the waiting lists are long; so, the proper targeting and management of care provision becomes more paramount [61,62]. Typical barriers to its implementation may include a lack of awareness of such potential, a lack of budget and commitment, and the absence of expertise and support, among others. The current review aims to promote and begins such recognition.

Questions on the cross-cultural adaptability of the reviewed tools are also of note [63,64]. All the measurements were developed and/or field-tested in North America, the United Kingdom, or Europe. Some tools, like the CPM, include specific domains designed to be generalizable in Western medical settings (e.g., PHQ-9 scores [48] the HoNOS [49]). Other tools have domains that leave room for broader interpretation; for example, the INTERMED has been adapted in many different cultural contexts and settings. Further cross-cultural validity research of the instruments is needed, particularly for mental health settings where the cultural interpretation and idioms of stress and illness explanatory models vary considerably from culture to culture [65]. The concept of complexity is known to be deeply and historically embedded in cultural epistemologies of health, wellbeing, and illness. A notable example is the Medicine Wheel, which Indigenous Peoples in North America use to conceptualize the complex interactions between the physical, mental, emotional, and spiritual domains [66]. In an modern example, the “Aaniish Naa Gegii” tool (Ojibwe for “How are you?”, also known as the Aboriginal Children’s Health and Wellbeing Measure (ACHWM)) comprises sixty-two questions across the four aspects of health represented by the Medicine Wheel [67]. The tool is used to assess the health and wellbeing needs of youths and to evaluate program and service delivery. It is important to recognize and study such non-Western approaches to complexity, as they provide valuable and complementary insights into advancing complexity science.

Lastly, from an end-user choice perspective, the current review notes the heterogeneity of the tools, which highlight all the key health system functions—across macro- (system and policy), meso- (organizational), and micro- (clinical) levels. No single tool fits neatly in any particular level, but for those interested in a macro-level tool, the VCAT is the most comprehensive and digitally advanced, developed to serve from an individual level up to population health management (PHM) and quality-of-care improvement at a system level. Furthermore, the VCAT was specifically designed for vulnerable patient populations with complex psychosocial and severe mental health and addiction needs [30]. Its drawback is the reliance on a multi-sourced database. It is most ready for the increasingly computer and large data-driven approach in formalizing a patient’s case conceptualization [68]. The INTERMED, on the other hand, with its ease of use and demonstrated success in being adapted and implemented in many settings and contexts, is useful at a meso- and micro-levels [44]. The INTERMED tool is suited for improving functions relating to organizing integrated healthcare for highly complex populations, including those with chronic psychiatric disorders, substance abuse, pain, and unexplained medical complaints [69]. Of note, both the VCAT and INTERMED were found to be well-suited for use in individuals with SMI at any level given their comprehensive and multi-domain approaches. The MCAM [37] and PCAM tools [38], both derived from the INTERMED tool, can be recommended for micro- and some meso-level use, as they are both best researched for improving primary care interdisciplinary teamwork and referrals to social services. Furthermore, the AMPS [36] tool could be considered for micro-level use, as it was specifically designed for complex populations, focusing on functions related to individualized care planning and outcome assessment.

This current review has several limitations. One is related to the classical problem within psychiatry in terms of a lack of biological markers to determine the diagnostic accuracy of the chief diagnosis, introducing potential complexity in and of itself. There was also a limited number of results returned from our academic search strategy. Trade and gray literature, or modified versions of the current tools, or media reports of successful implementations were not searched for or reviewed, and some suitable tools may have been missed. This was remediated to a degree by our hand search strategy, which reviewed all tools featured in the major review by Schaink and colleagues [1] and another major report from Shukor and colleagues [30]. Another limitation is related to the fact that addictions were not specifically included in the current review search terms, and addiction was not examined as a separate entity. The review does capture substance abuse issues; since addictions are often prevalent as comorbidities in people with mental illness, some tools have addiction-relevant domains or items. Despite efforts to locate and test-run the different tools, information on utility metrics, such as ease of use, time taken to complete, ideal databases needed, adaptability, and availability and accessibility for users, are not robust. Part of this reflects that the developers of the tools did not report such useful information in academic journals.

Based on the current review, some reflections on the future directions of the field are of interest. One relates to the general neglect of the patient perspective. Most of the tools are limited by the lack of input from the patients themselves, which is a major component of any reliable and valid patient complexity tool. This lack also reflects the general state of healthcare, where subjective information is often missing in health data or considerations [70,71]. Some tools integrate the data sources that currently exist (e.g., the VCAT), which means that it might be easier to include existing or future patient self-reported data. Another future improvement may be working towards more of an evidence-based consensus on the definition of complexity. The current divergent definitions may reduce user confidence in adapting such tools. There has been more convergence in the field on agreeing that complexity extends beyond the medical model. Focusing on higher data quality is another desirable direction as the quality of the complexity tool is only as good as the quality, rigor, and validity of the data informing it. Starting to use complexity measures where one can, using the data that are available, paying attention to missing and gaps in the data, and striving for better data may drive a better understanding and appreciation of the different domains of complexity, and this may in turn improve patient care overall.

## 5. Implications for Behavioral Health

This scoping review provides practical information on the concept and current state of patient complexity measurement for, potentially, a wide range of stakeholders in mental health, from clinicians to health system managers and administrators. The review provides key insights on strengths, uniqueness, and limitations through the analysis and comparisons of a concrete number of readily available tools for those interested or curious about using them. Complexity measurements—starting with diagnostic group identification—could help mental health services to leverage what data they have; be guided purposively to think and collect the key data needed to measure complexity and generate a comprehensive understanding of a patient’s current clinical picture, special considerations, challenges, and needs; and be better guided to meet the services needs and improve care. Faced with long waiting times, inappropriate matching of services, and often fraught doctor–patient relationships, complexity measurements could greatly aid in proper admission to matched services and gauge the duration, level, and intensity of services to provide for and assess the readiness to transition to other services or graduate from more intensive services, creating better transparency, clarity on guidelines, objectivity in decision making, and overall enhancing service use and outcomes in mental healthcare settings. Where traditional disease-focused management fails in terms of paying attention to complexity, the current review promotes the utilization of tools that are biopsychosocially oriented and more accurately understand and serve those in need.

## Figures and Tables

**Table 2 diagnostics-14-01300-t002:** (**a**) Tool domains and sources—scoring, outcomes, and recommendations. (**b**) Tool domains and sources—domains and sources.

(a)							
Tool	Scoring, Outcomes, and Recommendations						
	Outcome Assessed in Relation to Complexity Measured by the Tool		Outcome Scoring	Summary Score		Recommendations Provided on Scoring	
			*How is it measured (i.e., Likert scale)?*	*Can responses be summed? Are there sub-scores?*		*Does the tool scoring/output(s) include recommendations for practitioners?*	
**CPM**	Mental Health Complexity: Mild (routine care); Moderate (collaborative care); High (enhanced care).		Mild, moderate, and high have criteria for specific data sources.	Yes—Sub-scores are aggregated into overall complexity using their algorithm. Some sub-score classification criteria, such as those for the Patient Health Questionnaire, are drawn from published validated instruments. Other sub-score criteria, such as those for chronic pain severity, are based on the experience of Intermountain clinicians and researchers.		Mild = routine, primary care provider-based. Moderate = adds care manager and mental health specialist participation. High = increases care manager and mental health specialist participation.	
**INTERMED**	For assessing biopsychosocial case complexity in general healthcare, for the comprehensive assessment and treatment of a complex patient.		Scoring of the variables is universal: 0—no vulnerability/need. 1—mild vulnerability/need for monitoring or prevention. 2—moderate vulnerability/need for treatment or inclusion in treatment plan. 3—severe vulnerability/need for immediate consideration or intensive treatment. For each variable, anchors were defined to facilitate scoring.	Yes—The scores on the individual variables are summed, leading to a total score in the range of 0–60. High complexity: >20 for the total score. Low complexity: ≤20 for the total score.		High complexity indicates a higher need for care.	
**Escalation Tool**	Recommended response pathways developed by the Clinical Liaison Psychiatry (CLP) clinician and with managerial feedback.		Patients with risk factors across three or more domains, or with one or more key risk factors, were considered most likely to benefit from escalation.	No—The CLP team decided not to sum risk factors to form a total score with a cut-off threshold, as this may mask the recognition of one key factor driving complexity.		Regular meeting with the treating team regarding the patient (weekly minimum)—addressing identified risk factors. If there is no consensus or the conflict/problem persists. Includes: Considers involving hospital executives; Considers involving mental health executives; Considers involving clinical ethics service. Considers involving the hospital medical–legal team.	
**VCAT**	Complexity scores (weighted and unweighted).		Complexity scores were calculated for each domain (“Q-scores”) using a Likert-type scale (0–4). Q-scores were used to calculate a Composite Complexity Score (CCS).	Yes—overall total. Unweighted and weighted CCS in the range of 0–4 are available.		CCS and sub-domain scoring available do not have specific recommendations or action items attached to the scores. However, the CCS was used to identify existing clients who were and were not meeting the mandate of the service. Vancouver Coastal Health (VCH) is also leveraging the complexity scores to operationalize the fundamental building blocks of empanelment and team-based care within the CHCs.	
**AMPS**	Degree/level of complexity as a rating of 0–3, corresponding to the level of action needed.		Each item in the tool is scored using a scale ranging from 0 to 3, where 0 indicates “no complexity” and 3 indicates “very complex”.	Yes—overall total; total score out of 33 is calculated.		Level of action needed: No complexity = No concerns; Mildly complex = Easily managed with ongoing care; watch/prevent—explore interacting issues; Moderately complex = Form a well-integrated/multi-faceted plan and set it in motion (usually with some kind of team); Very complex = Immediate, intensive, and integrated action may be needed.	
**Complexity checklist**	Outcome is the selection of one or two reasons for why the patient is complex.		Checklist where a PCP selects all categories that apply to the patient and then ranks the top three issues they believe contribute most to the patient’s complexity.	No		Reasons for why a patient is complex: 1. Routinely requires more clinician time and resources than is normally allocated in the PCP’s practice, and/or 2. Fails to achieve satisfactory clinical outcomes due to his/her inability to adhere to PCP counsel.	
**Tool**	**Mental Health Diagnoses Content/Domain?**	**Exact Label/Title for Mental Health Component/Domain**	**Mental Health Domain Data Source**	**Social Determinants/Social Circumstances Domain?**	**Social Determinants Domain Data Sources**	**Additional Domains**	**Additional Domain Sources**
	*Is there a specific mental health domain?*			*Is there a specific social determinants domain?*			
**CPM**	Yes	1. Suicide Assessment. 2. Patient Health Questionnaire-9 (PHQ-9). 3. Anxiety/Stress Disorders. 4. Mood Disorder Questionnaire (MDQ). 5. Mood Regulation and ADHD subs-scores all within the objective category.	1. PHQ-9 response to Suicide State, Suicide Risk. 2. PHQ-9 Symptom Count, Severity Score. 3. Generalized Anxiety Disorder—7 Q1 Score, Q2–5 Score, and Q6–7 Score. 4. MDQ Q1, Q2, Q3 responses. 5. Adult ADHD Self-Report Scale (ASRS) Version 1.1 Part A Score.	None	None	Subjective Category: Number of Somatic Complaints, Chronic Pain Severity, Sleep Problem Severity, Substance Use Overall Impairment, Overall Health sub-domains. Objective Category: Family Relational Style, Family Pattern Profile, Most Common Support, etc.	Hospital admissions records and available data; patient-reported info used as a screening mechanism (with 47 pieces of data on 21 facets related to mental health).
**INTERMED**	Yes	1. Psychological variable (includes restrictions in coping and psychiatric dysfunction history, resistance to treatment, and psychiatric symptoms).	Tool scored from clinician’s perspective; information from the client based on a semi-structured interview.	1. Social.	Tool scored from clinician’s perspective; information from the client based on a semi-structured interview.	1. Biological. 2. Healthcare.	Tool scored from clinician’s perspective; information from the client based on a semi-structured interview.
**Escalation Tool**	Yes	1. Psychological (sub-domains include poor coping, psychiatric dysfunction/symptoms, treatment resistance, engagement, and readiness for change).	Tool scored from the clinician’s perspective.	1. Social (sub-domains include limited integration, social dysfunction, unstable housing, and restricted network).	Tool scored from the clinician’s perspective.	Biological, healthcare, previous complexity or escalation, and cognitive impairment.	Tool scored from the clinician’s perspective.
**VCAT**	Yes	1. Psychosocial factors domain (Q4). 2. Risk of harm to self or others (Q9).	1 (Q4): PARISProfile EMR Latest HoNOS Assessment (Q4 for cognitive, Q1 and Q8 for behavioral, and Q5 for functional impairment). 2 (Q9): IntraHealth Profile EMR Alerts (violence) PHQ-9 Extended leave PARIS EMR Extended Leave Alerts (violence) HoNOS Assessment (Q1 and Q2).	1. Social and environmental factors (Q3).	1 (Q3): PARIS EMR Latest HoNOS Assessment: question 11 for housing instability and question 12 for problems with occupation and activities IntraHealth Profile EMR Persons With Disabilities (PWD) forms Social History (SHX) codes.	1. Attachment (Q1). 2. Service density (Q2). 3. Relationships (Q5). 4. Activities of daily living (Q6). 5. Medical complexity (Q7). 6. Acute (hospital) utilization (Q8).	1 (Q1): IntraHealth Profile EMR Encounters PARIS EMR Encounters. 2 (Q2): IntraHealth Profile EMR Encounters PARIS EMR Encounters Referrals to services. 3 (Q5): PARIS EMR Latest HoNOS Assessment (Q9, Q11 and Q12) IntraHealth Profile EMR SHX codes. 4 (Q6): PARIS EMR InterRAI-MDS assessment in Home Health (MAPLE scores, CAPS) Occupational Therapy (OT)/Physiotherapy (PT) assessments for mobility Latest HoNOS Assessment (Q5 for physical illness and disability, Q10 for activities of daily living, Q11 for housing, and Q12 for occupation and activities). 5 (Q7): IntraHealth Profile EMR Problem List Medications (EMR) PSW forms SHX codes PARIS EMR Latest HoNOS Assessment (Q6, Q7, and Q8 for mental health issues, and Q3 for substance misuse). 6 (Q8): EDMart and AcuteMart ED visits by CTAS LOS (acute admissions). Q9: IntraHealth Profile EMR Alerts (violence) PHQ Extended leave PARIS EMR Extended Leave Alerts (violence) HoNOS Assessment (Q1 and Q2).
**AMPS**	Yes	1. Psychiatric (general assessment, mental health, and addictions).	Tool scored from the clinician’s perspective and any information obtained directly from the client could be used to inform their assessment.	1. Social (includes housing, poverty, social support, and readiness for change).	Tool scored from the clinician’s perspective and any information obtained directly from the client could be used to inform their assessment.	1. Attachment (ongoing relationship with GP or not). 2. Medical (severity of symptoms and challenges with the management of medical problem(s)).	Tool scored from the clinician’s perspective and any information obtained directly from the client could be used to inform their assessment.
**Complexity checklist**	Yes	1. Mental or emotional health problems.	Tool scored from the clinician’s perspective.	None.	None.	1. Multiple clinical diagnoses. 2. Lack of patient self-activation. 3. Insurance/financial issues. 4. Problems with navigating the healthcare system. 5. Frequent admission to the emergency room, urgent care, or hospital. 6. Family or relationship difficulties. 7. Cultural issues or language problems. 8. Patient literacy or educational limitations. 9. Limitations due to patient’s cognitive functioning. 10. Lack of trust in medical providers. 11. Other issues (please indicate). 12. Number of active diagnoses that you currently manage for this patient. 13. Lack of social systems support.	Tool scored from the clinician’s perspective.

**Table 3 diagnostics-14-01300-t003:** Utility of tools for practitioners.

Tool	Ease of Use	Mode of Use	Time Taken to Complete	Adaptability	Recommended by Government or National/International Association?	Patient Specificity, Subspecialty, and/or Recommended Settings	Available Languages	License Required for Use?
		*(Pencil and paper versus computer-administered)*	*(Minutes, if given, or number of indicators as a proxy)*	*(Is partial completion and scoring still valid?)*		*(i.e., Adolescent, geriatric, disability, etc.).*		*(Cost)*
**CPM**	Moderate	Originally, the CPM was a packet provided to the patient to fill out. This paper details the process of the computerization of the data in a pilot.	The packet solicits 47 pieces of data from the patient on 21 facets related to mental health (some facets involve more than 1 piece of data).	Uncertain.	No information.	Patients screened at Intermountain Healthcare’s Primary Care Clinical Program.	English.	Not applicable.
**INTERMED**	Studies conducted during the last 10 years show that the INTERMED has face validity, is brief and easy to use, and is reliable and valid.	Pencil and paper.	Healthcare professionals need about 20 min for the interview.	Yes.	No information.	Has been tested in somatic populations, mental health settings, and in a wide range of populations, such as diabetes, low back pain, and multiple sclerosis.	The INTERMED Self-Assessment is translated in 9 languages and developed for specific groups of caretakers.	Uncertain.
**Escalation Tool**	Clinicians judged it to be a useful and objective way of operationalizing complexity. The tool was considered quick, easy to use, and stimulated thought.	Pencil and paper.	The checklist has 6 domains and between 2 and 5 risk factors in each domain.	Uncertain.	No information.	Consulting Liaison Psychiatry.	English.	Free within the article.
**VCAT**	Producing outputs appears easy, although the integration of the tool into a health system would require some effort.	Computer.	Domain data are drawn from existing sources, meaning the VCAT-CM algorithm can be updated monthly (currently).	Yes—Ease of adjusting complexity domain weightings to suit local contexts, values, and perceptions is a key strength of the VCAT-CM.	No information. Note: Some domain-specific sources, like the HoNOS and PHQ-9, are recommended for use by international associations.	Highly complex and marginalized population accessing CHCs.	English.	Not applicable.
**AMPS**	Uncertain.	Pencil and paper.	Ideally completed within 10 min, or less once the practitioner is more familiar with the tool * * Author communication.	Uncertain.	No information.	Provides primary care services to individuals aged 19+ years who do not have a regular family doctor (general practitioner or GP) or nurse practitioner (NP) and face complex medical, mental health, addiction, and/or socioeconomic needs.	English.	Free within the article.
**Complexity checklist**	PCPs found the screen easy to use and feasible to integrate into their routine practices.	Pencil and paper.	Participating physicians reported the average time to complete the screen was 1.4 (+/− 0.6) minutes.	Uncertain.	No information.	General PCPs.	English.	Free within the article.

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
