# Peer review of "Beyond Diagnosis and Comorbidities—A Scoping Review of the Best Tools to Measure Complexity for Populations with Mental Illness"

_diagnostics, 2024, doi:10.3390/diagnostics14121300_

Round 1
Reviewer 1 Report
Comments and Suggestions for Authors
This review is generally well written and provides much valuable information on this emerging literature involving the assessment and measurement of complexity in mental health. It provides a rich description of 14 of the most promising instruments, along with some reasonable recommendations regarding their use. The following is what I would consider the most pertinent limitations of the review:
1. Regarding the review of the literature, the authors are correct in highlighting the heterogeneity of the definitions and methods used in this area. But, I was disappointed that so little, if anything, was covered concerning the contributions of the actual literature on complexity theory. Most all of the definitions used generally involved simply how complicated or numerous some patients' problems and diagnoses are. Whereas in complexity theory considerable attention is paid to nonlinearity, interactive causal structures, chaos theory, and especially "edge of chaos" phenomenon and its role in adaptiveness and resilience. There are many reviews of this literature, for example: Hudson, C.G. (2009). Complex Systems and Human Behavior. Oxford. Also, a sharper distinction and discussion of complex systems in mental illness versus mental health might be included, although this is alluded to in places.
2. In both the narrative and various tables that describe the instruments, there is remarkably little in the way of psychometric indicators that may possibly have been reported on the reliability and validity of the instruments reviewed, interpretative norms, and populations on which they may have been tested and normed on. If reported, these should be summarized, and if none, then this should also be noted and discussed as a significant limitation of the development of the various instruments.
3. The extensive detail and layout of the tables don't seem to work very well and are hard to follow. This may be somewhat ameliorated once committed to print in smaller fonts. However, the authors should consider either much condensed tables, and/or more detailed tables presented in appendices to the article.
But, on the whole, this is an important paper that will be of much value to many readers.
Comments on the Quality of English LanguageGenerally the English is fine with the exception of some occasional typos, i.e. :
Line 114 ("base don")
Line 156 (capitalization of "Tables")
Author Response
Please see the attachment that addresses comments from both reviewers.

Reviewer 2 Report
Comments and Suggestions for Authors
This was unlike many manuscripts that I have reviewed. It was a fairly comprehensive method of determining domains and time perspectives for clinicians active in mental health evaluations. I am not sure that I am inclined to think that this is "journal ready", I definitely COULD see it having value to the academic community, which is why I would only suggest minor changes. Along those lines, please do the following:
1) Please employ the services of a copy editor. There were more than a few grammatical and spelling errors that are worthy of attention.
2) Please see about cleaning up Table 1. While it contained good information, there were awkward breaks in words that distracts from the quality of the table.
3) I thought that the discussion and limitations section seemed on point, so my thanks to the author team for that.
Author Response
Please see the attachment that addresses comments from both Reviewers

Round 2
Reviewer 1 Report
Comments and Suggestions for Authors
This paper is excellent, with the critiques substantially and effectively addressed.